# Advancing workpiece dimension measurement: Integrating AI-based edge detection with machine vision and coordinate measuring systems

Yazid Saif[1]*, Anika Zafiah M. Rus[1]*, Yusri Yusof[1], Yeong Hyeon Gu[2]*,
Mohammed A. Al-masni[2], Shehab Abdulhabib Alzaeemi[3], Osamah Al-qershi[4],
Yahya M. Altharan[1], Sami Al-Alimi[5]

**1** Advanced Manufacturing and Materials Center (AMMC), Faculty of Mechanical and Manufacturing, Engineering, Universiti Tun Hussein Onn Malaysia (UTHM), Batu Pahat, Johor, Malaysia, **2** Department of Artificial Intelligence, College of Software and Convergence Technology, Sejong University, Seoul, Republic of Korea, **3** IRC for Finance and Digital Economy, KFUPM Business School, King Fahd University of Petroleum & Minerals, Dhahran, Saudi Arabia, **4** Centre for Epidemiology & Biostatistics, Melbourne School of Population and Global Health, University of Melbourne, Melbourne, Australia, **5** Faculty of Mechanical Engineering Universiti Teknologi MARA (UiTM), Shah Alam, Selangor, Malaysia

* yazeedalkosa@gmail.com (YS); zafiah@uthm.edu.my (AZMR); yhgu@sejong.ac.kr (YHG)

## Abstract

Image preprocessing and edge detection are critical in industrial machine vision for workpiece dimension measurement. Challenges arise from interference regions on workpiece surfaces, complicating edge detection and roundness assessment. This paper investigates the application of AI-based detection methods within the industrial image analysis framework of coordinate measuring machines. Initially, two models with varying hole sizes and counts were designed in SolidWorks, fabricated using a Prolight 3-axis CNC milling machine, and analyzed. A transfer learning approach mitigated overfitting on the limited dataset of model surface features. The study employed a Convolutional Neural Network (CNN) to identify interference regions and predict circularity, enhancing measurement accuracy. Validated with a testing dataset, the CNN achieved 100% classification accuracy, confirmed by a Confusion Matrix. Fine-tuning of the CNN with specific training data leveraged image preprocessing to enhance features via multi-layer convolution, pooling, and detailed analysis through fully connected layers. Comparative diameter analysis across Models 1–2 showed all methods maintained ≤0.05 mm deviation from actual values, with CNN exhibiting minor variations at Model 1's points 3,7,9 while matching CMM precision (r = 1.000) and outperforming vision systems in Model 2's multi-hole measurements, supported by ANOVA-confirmed discrimination (F = 34,514,683, p < .001) and cross-material scalability to Drelin via 200-image retraining. The results underscore the effectiveness of integrating deep learning techniques into industrial inspection, contributing a standardized methodology for precise workpiece dimension measurement. This research highlights the potential of combining machine vision, deep

**Data availability statement:** All relevant data are within the manuscript and its Supporting Information files.

**Funding:** This work was supported by Institute for Information & Communications Technology Planning & Evaluation (IITP) grant funded by the Korea government (MSIT) (No. RS-2025-25441838, Development of a human foundation model for human-centric universal artificial intelligence and training of personnel).

**Competing interests:** The authors declare that they have no known competing financial interests or personal relationships that could have appeared to influence the work reported in this article.

learning, and coordinate measuring systems to advance industrial measurement processes.

## Introduction

Significant advancements in computer-aided design and analysis have revolutionized signal processing and simulation testing in the engineering industry. Conventionally, manual inspection was one of the most labour-intensive and inefficient processes, often leading to human errors such as fatigue and mismeasurement, which could reduce product quality. To mitigate these issues, automatic optical inspection systems have become increasingly widespread. Hardware and software advancements have recently been aided by Artificial Intelligence (AI) [1], image processing [2], computer vision [3], and deep learning [4]. Significant industrial needs continue despite increasing research on AI-driven inspection systems and machine vision: (i) robust detection of difficult interference regions in a variety of mechanical and lighting conditions; (ii) automated feature extraction and selection that generalizes across different workpiece materials and geometric features; and (iii) real-time scalability to match Industry 4.0 production lines [1,5]. The integration of CNN with coordinate measuring machines (CMM) and traditional vision systems in a single pipeline is still a significant gap, although numerous research have employed CNN-based techniques for defect identification [6]. Closing this gap could help reduce production delays and improve dimensional accuracy by combining high-throughput non-contact sensing (vision systems and deep learning) with offline, contact-based precision measurements (CMM).

Particularly, integrating deep learning models with existing optical inspection systems through image processing and deep learning detection is growing in popularity. This breakthrough addresses bottlenecks in manufacturing defect detection and enhances the manufacturing process. Hence, in industrial production, quality control plays a critical role in ensuring products meet delivery requirements. Industrial measurement, encompassing online and offline methods, is pivotal in this process. Online measurement, distinguished by its automation, high reliability, precision, and minimal human influence, outperforms offline methods. Despite its stability and resistance to interference, contact measurement is time-consuming, affecting processing cycles and limiting point-specific measurements. Non-contact methods such as laser measurement [7] and machine vision [8,9] have gained prominence due to their broad measurement range, speed, and cost-effectiveness.

### Previous works

The process of laser measurement necessitates the scanning of the entire surface to obtain profile information, a procedure that is inherently time-consuming. On the contrary, machine vision employs cameras to capture surface data instantaneously, thereby ensuring superior speed and precision for online applications. Traditional machine vision techniques typically involve edge detection post-image preprocessing, translating edge pixel distances into physical measurements [10,11], or workpiece

surface roughness [12]. For instance, [11] developed an online machine vision system for measuring rubber extrusions with measurement uncertainty constrained to 0.1 mm. It introduced a machine vision algorithm that successfully detects machine tool wear with a 6.7% error compared to manual methods [13]. Image-based cross-correlation analysis was utilised for microscale machine tool wear detection, with a detection capability down to 0.1 mm [14]. High-quality original images and controlled detection environments are crucial for reliable measurements yet achieving these in industrial settings poses challenges due to harsh conditions. Two main challenges must be addressed for adequate industrial machine vision: managing interference regions like cutting fluid and chips on workpiece surfaces, which can create noisy edges, and accurately detecting edges amidst textured surfaces post-roughing. Interference region detection, akin to defect identification, often employs thresholding methods; Otsu's automatic thresholding method addresses fixed threshold instability [15]. Additionally, existing approaches often rely on carefully controlled illumination or simple thresholding methods, which are unreliable in real-world conditions [11,16]. Although advancements, detecting interference regions remains complex in dynamic processing environments. Moreover, traditional edge detection methods exhibit poor robustness and parameter sensitivity [17] due to inadequate feature differentiation in industrial contexts. Practical feature engineering is pivotal in traditional machine vision, with feature selection and extraction as core challenges. Developing a generalized online vision measurement approach for workpiece dimensions that automates feature selection is essential for advancing industrial machine vision capabilities.

AI directly learns the mapping between inputs and targets, removing the requirement for preparing input images compared to past approaches. Studies highlight AI's superiority in image processing [4], with Support Vector Machine (SVM) [18], Neural Networks (NN) [19], and other algorithms that are gradually replacing manual feature extraction. Although the SVM excels in classification tasks like character recognition, face detection, and text categorization, it is less suitable for detecting interference regions and extracting edge images due to high computational costs and slow speeds, as NNs increasingly outperform SVM. Early NN structures struggled with complex image tasks, but Convolutional Neural Networks (CNNs) have since surpassed traditional methods and other deep models in various computer vision applications [20]. However, practical manufacturing adoption requires comprehensive solutions combining CNN-based detection, contact-based precision from a CMM, and standard machine vision strategies. This paper introduced a small data-driven convolution neural network (SDD-CNN) for roller minor defect inspection, utilizing an ensemble strategy for tiny data preprocessing [6]. CNN remains pivotal in Vision-Based Measurement (VBM) systems, dominating deep learning techniques in this field [21]. On the contrary, Deep learning's image classification technology offers a novel approach to image detection, enhancing detection accuracy for surface inspections [22,23]. The study investigates deep learning algorithms, their evolution, computational methodologies, emerging research areas, and future trends in smart manufacturing, focusing on enhancing system performance and decision-making [5]. Advancements in graphics processing units have significantly boosted hardware computing capabilities. Leading frameworks like TensorFlow [24] and PyTorch [25] have played crucial roles in advancing deep learning technologies. YOLO [26], a prominent framework, is widely utilized for object detection and image processing. Numerous studies employ convolutional neural network models to classify diverse images [27], leveraging their ability to automatically extract relevant features [28,29]. Furthermore, the LeNet-5, introduced by Lecun et al., marked a significant advance in handwritten character recognition, laying the foundation for subsequent CNN advancements [30]. Subsequent CNN models, such as AlexNet [31], GoogleNet [32], and ResNet [33] have progressively enhanced classification accuracy on ImageNet [34], driving extensive research and application in object detection, semantic segmentation, and other image-processing domains [35]. Convolutional neural networks (CNNs), in particular, have seen remarkable performance gains in object detection and classification of images since artificial neural networks were addressed [36].

The CNN has performed exceptionally on diverse datasets, including ImageNet and CIFAR-100 [37]. Therefore, these application cases demonstrate the importance of using CNNs for industrial sector measurement. Convolutional neural networks, or CNNs, are perfect for interference and edge detection in industrial detection. Huang et al. developed automated

 

defect inspection systems for high-speed trains based on CNN and intelligent damage detection techniques for steel wire ropes using convolutional neural networks, achieving 0.99 percent accuracy. [38]. Zhang et al. establish a vision-based fusion approach for detecting defects in milling cutter spiral cutting edges, reaching 98.86% accuracy [39]. This study addresses the critical challenges in industrial measurement by integrating CNNs with machine vision systems and CMM to develop a comprehensive methodology for workpiece dimension analysis. The proposed approach leverages CNNs to enhance edge detection and mitigate the impact of surface interference, enabling precise circularity and dimensional analysis. Furthermore, this research introduces a deep transfer learning framework tailored for Delrin workpiece images, optimizing classification accuracy with a compact and efficient CNN architecture. Nevertheless, the proposed models employ their learned weights on the ImageNet dataset in conjunction with a CNN structure to extract features. Extensive experiments are conducted to assess the effectiveness of the proposed model on Delrin workpiece images captured from the vision system. Additionally, comparisons are made by evaluating various widely recognized indicators based on the confusion matrix. The key contributions of this work include:

- The development of a CNN-based model for detecting and repairing interference regions in workpiece images,

- A robust method for edge detection and hole measurement leveraging deep learning, and

- A comparative evaluation of CNN predictions against CMM and vision systems to validate measurement accuracy.

By addressing existing gaps in precision and robustness, this study contributes to the advancement of industrial metrology and underscores the potential of deep learning technologies in transforming quality control practices. The findings not only improve dimensional measurement accuracy but also establish a foundation for future exploration of AI-driven inspection systems in manufacturing, aligning with the principles of Industry 4.0 and intelligent automation.

This paper proposes a CNN-based methodology that detects and corrects interference regions in workpiece images, enhancing sub-pixel edge detection and roundness measurement. We compare the CNN results with data from a coordinate measuring machine (CMM) and a vision system to demonstrate reliability. We further highlight how the approach can generalize beyond Delrin workpieces with multiple-hole geometries, emphasizing that it can handle materials of different textures and shapes with minimal retraining or transfer learning. The results underscore the system's potential to reduce measurement time and improve dimensional accuracy—a meaningful step toward Industry 4.0 smart manufacturing [21,40].

This article are structured as follows: **Section 2:** The previous work presenting on roundness measuring, including AI and CNN. **Section 3: The Methods** describes the materials, experimental setup, and planned process. **Section 4:** The presenting and discussing the findings. **Section 5:** Draws the conclusion and future trends.

## Materials and experimental setup

### Materials

(Fig 1) shows the Computer-Aided Design (CAD) models used to assess the surface properties of the side view layout of two models. Model 1 has a single hole, whereas Model 2 has five holes that require examination. The inspection circles occupy most of the model's surface area, allowing for the detection and measurement of the outer edge surface. Consequently, the machining parameters for mild DELRIN workpieces with round holes are specified as follows: Model 1 has a hole diameter of 30 millimetres (mm), while Model 2 includes hole diameters of 30, 28, 26, 23, and 20 mm. The parameters for processing both models are 2000 revolutions per minute (rpm) for the spindle, 0.1 mm/rev for the feed rate, 20 mm for the cutting depth, and 250 m/s for the cutting speed. In this study, an artificial intelligence approach was implemented for the detection and prediction of holes in models from images that were reconstructed using CNN.

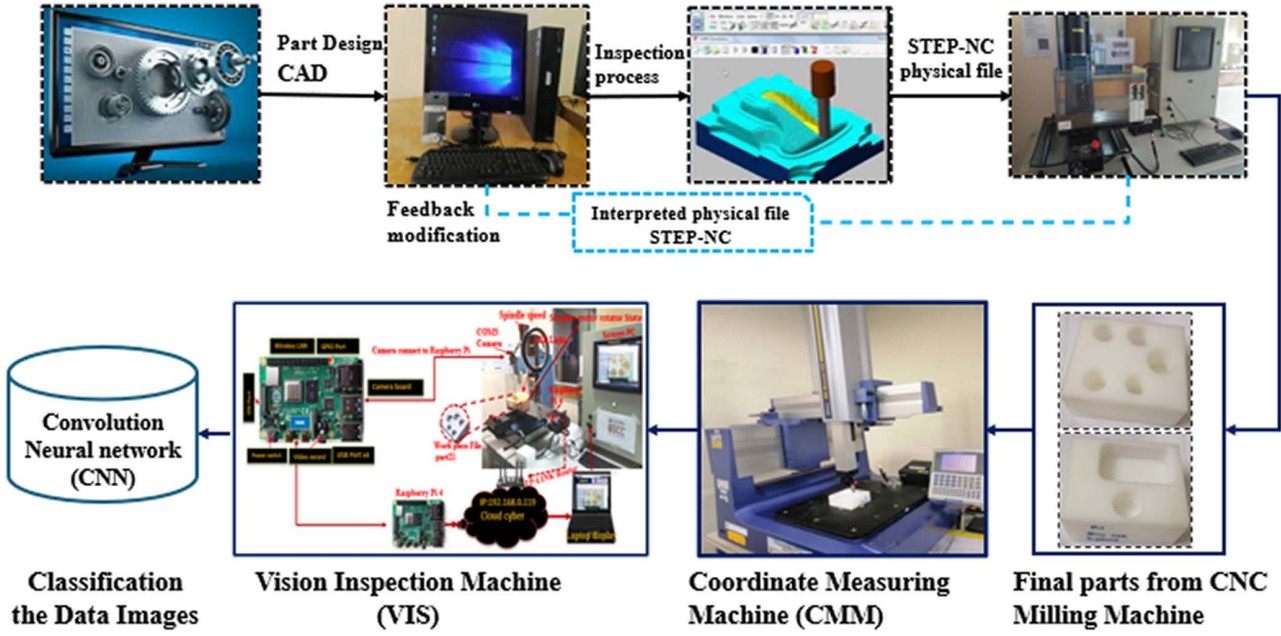

**Fig 1. Implementing the automation system measures data through VIS, CMM, and CNN.**

## Experimental setup and image acquisition

The experimental setup integrates hardware and software to perform the model 1 and 2 tests using a vision system. Based on the optimal lighting is crucial for capturing high-quality images of the test objects. A servo motor rotates the workpiece, and an encoder on a fixed axis measures the rotation angle. This system was implemented on an INTELITEK PROLIGHT three-axis milling machine, adhering to Example 1, Part 21, of the ISO 14649 standard conducted by [40,41].

Concurrently, Image acquisition was a critical step, performed under controlled conditions to ensure dataset quality and representativeness. A COMS camera specification CMOS sensor, Global shutter Resolution: 640X480MJPEG 30fps and resolution, if available, should be added here, e.g., "a 5MP COMS camera") was used to capture visual data of the target area on the workpieces. Optimal and consistent lighting conditions were maintained using describe lighting setup, e.g., "a diffused LED ring light" to minimize shadows and ensure uniform illumination, which is crucial for capturing high-quality images suitable for subsequent analysis. The LED lighting emerges as more than just illumination; it assumes a role in signalling, status indication, and even data transmission. The workpiece was rotated by a servo motor, and an encoder on a fixed axis measured the rotation angle, allowing for images to be captured from multiple perspectives if required for the analysis as suggested through Table 1 for the vision system. Efforts were made to maintain consistent workpiece positioning relative to the camera for each image capture session. These controlled conditions aimed to reduce variability unrelated to the workpiece features themselves, thereby enhancing the rigor of the experimental data collection. The captured images formed the basis for the dataset used by the neural network to predict the circle value of each hole. The extended analysis of the models is based on comparing the data from this vision system, the CMM, and the predictions of the CNN.

**Implementation of models.** A specific system design for the INTELITEK PROLIGHT 3-axis milling machine was developed according to Example 1, Part 21 of the ISO 14649 standard (STEP-NC). The CAD model significantly aided the design process, particularly in identifying the surface region visible within the side view. This region corresponded to a specific hole and pocket targeted for inspection. While, the inspection strategy focuses on an "inspection circle" marking

**Table 1. Diameter values from the Vision system of Model 1 and Model 2.**

| Image No. | Angle in degree | Diameter of Model 1 | Diameter of Model 2 | | | | |
|---|---|---|---|---|---|---|---|
| | | 30mm | 20 mm | 23 mm | 26 mm | 28 mm | 30 mm |
| 1 | 0° | 29.9383 | 19.8432 | 23.5647 | 26.6002 | 27.9027 | 30.1693 |
| 2 | 31° | 29.9397 | 19.8412 | 23.5573 | 26.6101 | 27.9039 | 30.1673 |
| 3 | 62° | 29.9378 | 19.8421 | 23.5623 | 26.6018 | 27.9111 | 30.1695 |
| 4 | 93° | 29.9292 | 19.8409 | 23.5609 | 26.6105 | 27.9171 | 30.1681 |
| 5 | 124° | 29.9364 | 19.8361 | 23.5695 | 26.6038 | 27.9101 | 30.1598 |
| 6 | 155° | 29.9281 | 19.8452 | 23.5637 | 26.6031 | 27.9119 | 30.1691 |
| 7 | 186° | 29.9368 | 19.8432 | 23.5672 | 26.6106 | 27.9107 | 30.1706 |
| 8 | 217° | 29.9311 | 19.8382 | 23.5668 | 26.6061 | 27.9076 | 30.1679 |
| 9 | 248° | 29.9315 | 19.8407 | 23.5651 | 26.6064 | 27.9016 | 30.1659 |
| 10 | 279° | 29.9299 | 19.8412 | 23.5692 | 26.6074 | 27.9103 | 30.1697 |
| 11 | 310° | 29.9324 | 19.8422 | 23.5659 | 26.6029 | 27.9101 | 30.1689 |
| 12 | 360° | 29.9308 | 19.8439 | 23.5619 | 26.5995 | 27.9119 | 30.1692 |

the model's most critical feature: the outer edge surface. Positioning this circle ensures precise measurement of that edge.

Two distinct models provide the complete machining characteristics required for producing DELRIN prismatic workpieces on the PROLIGHT mill: As visually detailed in (Fig 1):

• Model 1: Features a single round hole with a 30-millimeter diameter.

• Model 2: Includes a series of holes with diameters of 30 mm, 28 mm, 26 mm, 23 mm, and 20 mm.

The machining process for these parts follows a precise set of specifications designed for efficiency and accuracy. Key parameters include a cutting width of 20 millimeters, a controlled feed rate set at 0.1 millimeters per revolution, a spindle speed of 2000 revolutions per minute (RPM), and a high cutting speed of 250 meters per second. These settings collectively optimize material removal while ensuring surface quality on the DELRIN workpieces. The design itself builds upon Example 1 from ISO 14649 Part 21 (STEP-NC), specifically extending it to accommodate the features of both models.

**Dataset of models.** The dataset comprised a training subset of 651 images and a testing subset of 114 images. This sample size, while modest, was deemed sufficient for initial validation of the proposed CNN approach for this specific workpiece geometry and interference types, aligning with similar exploratory studies in specialized machine vision applications where extensive datasets are not always readily available. Future work will aim to expand this dataset to ensure broader model generalizability. During the training process, fifteen percent of the training subset (approximately 97 images) was randomly chosen to serve as a validation subset, used to monitor for overfitting and guide hyperparameter tuning. Data augmentation included random horizontal and vertical flips, small random rotations (±5°), and minor brightness adjustments (±5%). These augmentations enhance model robustness to real-world variability [6].

## Inspection data with CMM

A Coordinate Measuring Machine (CMM) is a device utilized in manufacturing and engineering for tasks such as quality control, reverse engineering, component alignment, prototype inspection, and complex form measuring. In this study, the CMM (MITUTOYO QM-Measure 353) serves as the essential base actuality control for validating the accuracy of the Vision Inspection System (VIS) and the proposed CNN-based measurement method. Its established high precision and reliability in dimensional metrology make it an appropriate benchmark. The CMM quantifies

geometric characteristics and verifies that components meet design specifications. Furthermore, it quantifies geometric characteristics and verifies that components meet the design specifications. (Fig 2) depicts the sequential process of measuring the holes in the Derlin workpiece using a Coordinate Measuring Machine. Hence, an accurate device designed specifically for measuring hole roundness is the MITUTOYO QM-Measure 353. The device has a measurement range of 20 inches along the X-axis, 12 inches along the Y-axis, and an operational envelope measuring 20 inches in length, 12 inches in width, and 12 inches in height. The gadget has a resolution of 0.00002 inches, a maximum work height of 16.14 inches, and a maximum work weight of 66 pounds. The device's dimensions are as follows: length (L) = 33 inches, width (W) = 35 inches, and height (H) = 88 inches. The MITUTOYO QM-Measure Coordinate Measuring Machine (CMM) uses a contact probe to identify surface features, such as hole circularity. To ensure the reliability of these control measurements, each specified hole diameter on Model 1 and Model 2 (as reported in Table 2) was measured multiple times, and the average or a representative stable value was used, minimizing operator induced variability and confirming measurement consistency.

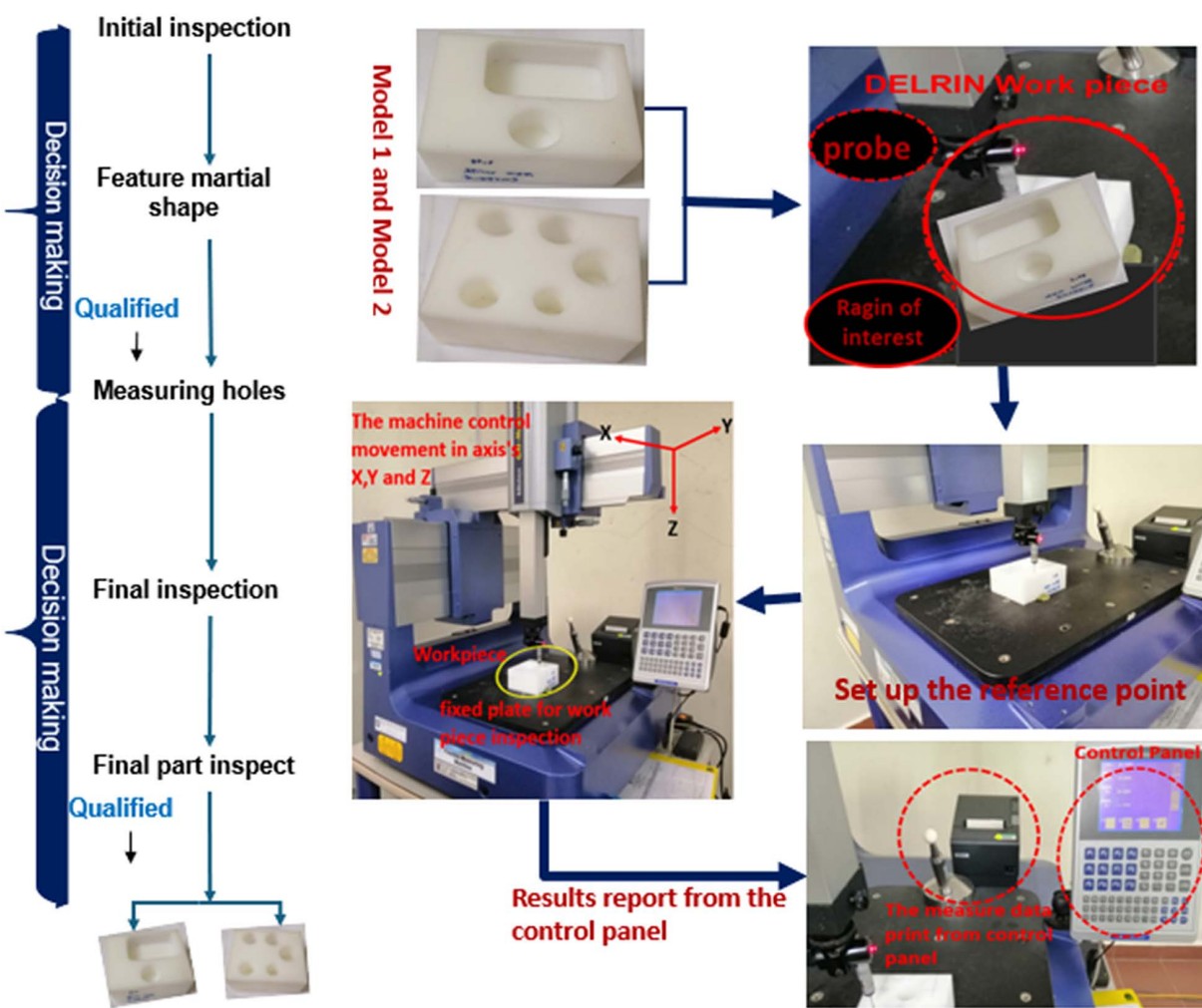

**Fig 2. The inspection procedure in CMM for Model 1 and Model 2.**

**Table 2. Diameter values of Model 1 and Model 2 from the CMM machine for the Delrin workpiece.**

| Image No. | Diameter Model 1 | Diameter Model 2 | | | | |
|---|---|---|---|---|---|---|
| | 30mm | 20mm | 23 mm | 26 mm | 28 mm | 30 mm |
| 1 | 30.0121 | 20.0097 | 23.0094 | 25.9917 | 27.9805 | 30.0219 |
| 2 | 30.0099 | 20.0061 | 23.0113 | 25.9921 | 27.9819 | 30.0206 |
| 3 | 30.0112 | 20.0089 | 23.0071 | 25.9911 | 27.9799 | 30.0201 |
| 4 | 30.0123 | 20.0094 | 23.0084 | 25.9928 | 27.9801 | 30.0199 |
| 5 | 30.0133 | 20.0098 | 23.0098 | 25.9897 | 27.9812 | 30.0196 |
| 6 | 30.0113 | 20.0095 | 23.0102 | 25.9909 | 27.9858 | 30.0195 |
| 7 | 30.0104 | 20.0074 | 23.0091 | 25.9913 | 27.9829 | 30.0185 |
| 8 | 30.0109 | 20.0087 | 23.0093 | 25.9922 | 27.9828 | 30.0175 |
| 9 | 30.0124 | 20.0091 | 23.0049 | 25.9923 | 27.9834 | 30.0169 |
| 10 | 30.0116 | 20.0065 | 23.0055 | 25.9925 | 27.9825 | 30.0165 |
| 11 | 30.0128 | 20.0069 | 23.0086 | 25.9915 | 27.9815 | 30.0159 |
| 12 | 30.0103 | 20.0072 | 23.0101 | 25.9902 | 27.9823 | 30.0157 |

## Vision inspection system (VIS)

The accuracy of the VIS system in measuring holes on a workpiece's surface relies heavily on selecting appropriate computer vision algorithms. These algorithms are designed to detect and analyze holes' unique characteristics, forming the measurement process's foundation. For example, edge detection algorithms are essential for precisely outlining the contours of holes, enabling accurate quantification.

## Deep learning classification model and transfer learning approach

This investigation utilizes an artificial intelligence framework, specifically Convolutional Neural Networks (CNNs), to detect and anticipate defects in model images, emphasising hole configurations. The specific CNN architecture employed builds upon a DenseNet-like structure [42], selected for its efficient feature reuse and gradient flow, which is particularly beneficial for learning from smaller datasets. A transfer learning strategy was adopted by initializing the network with weights pre-trained on the large-scale ImageNet dataset [34]. This approach helps to leverage generalized visual features learned from a diverse dataset, mitigating the risk of overfitting on our comparatively limited dataset of Delrin workpiece images and accelerating convergence. The research compares two distinct models: Model 1, which has a single hole with a diameter of 30 mm, and Model 2, which has five apertures with diameters of 30, 28, 26, 23, and 20 mm, all of which are subjected to the same cutting conditions. In addition to detecting defects, the CNN is employed to predict the circle value of each hole, further enhancing the precision of the analysis. CNNs were chosen for this project because of their proven ability to systemically extract relevant information from images, thereby enhancing the accuracy of defect identification during the manufacturing process. Although CNN can effectively distinguish between the two models, there is a thrilling opportunity to improve its classification capabilities further to identify more subtle differences in hole topologies. This potential emphasizes the CNNs in intricate industrial environments, where variations in hole dimensions and spatial configurations can enhance product quality. The research demonstrates that the robustness and precision of CNNs in identifying and forecasting errors across multiple contexts are improved by incorporating varied training data. Future research may investigate a wider range of manufacturing parameters and hole geometries, thereby offering additional opportunities to evaluate and improve the model's efficacy. This study validates the effectiveness of CNNs in identifying specific issues and provides a foundation for future research that will enhance the adaptability of these models for various industrial applications. Enhancements to quality control standards and more efficient manufacturing processes could result from advancements in this field, which would also address current challenges associated with dataset diversity and model performance.

 

Deep convolutional neural network architecture was introduced in 2017 by Huang et al. [43], which consists of 201 layers and an innovative dense connectivity pattern that promotes feature reuse and alleviates the vanishing gradient problem. Unlike traditional ConvNets, where each layer receives input only from the preceding one, the structure of CNN connects each layer to all preceding layers via concatenation (Eq. 1):

$$X_l = H_l([X_0, X_1, \ldots, X_{l-1}])$$ (1)

where $X_l$ is the output of the layer $l$, $H_l$ is the transformation function of the layer $l$, and $[X_0, X_1, \ldots, X_{l-1}]$ is the concatenation of the feature maps produced in layers $0, \ldots, l-1$.

This dense feature reuse encourages feature propagation across the network, allowing earlier layers to influence later ones even through multiple intermediate layers. Furthermore, CNN incorporates identity shortcut connections (Eq. 2), directly adding the input of a layer to its output:

$$X_l = F_l \cdot X_{l-1} + X_{l-1}$$ (2)

where $F_l$ is the transformation function of layer $l$ with bottleneck structure.

These shortcut connections bypass multiple layers, mitigating the vanishing gradient problem by directly transferring gradients to earlier layers during backpropagation. The CNN architecture effectively classifies pneumonia in chest X-rays, demonstrating high diagnostic accuracy and potential for clinical application due to their dense connections and shortcut paths [42,44]. The success of CNN lies in its ability to efficiently reuse features and improve information flow, paving the way for further advancements in image recognition and beyond.

### Training setup and hyperparameter rationale

A focused preprocessing procedure was implemented to balance the high resolution of our images with computational efficiency and manageable dimensionality. First, all images were resized to a standardized 224 x 224 pixels, aligning with the CNN input layer. Further, pixel intensities were normalized to the [0, 1] range, optimizing training efficiency while maintaining crucial image information. Next, we trained our proposed model using the Adaptive Moment Estimation (Adam) optimizer. The selection of hyperparameters a learning rate of 0.0001, a decay rate of 0.95, decay steps at 20 intervals, and a batch size of 2 was guided by a combination of common practices in similar CNN-based image classification tasks and preliminary empirical tuning on a subset of the validation data to achieve stable convergence and optimal performance. The model was trained for up to 30 epochs, with early stopping criteria monitored on the validation set to prevent overfitting, although for this dataset, training for the full 30 epochs yielded the best validation results. To ensure the robustness of the reported CNN performance (100% classification accuracy and MAE values), the entire training and testing process was repeated three independent times with different random seeds for data shuffling and weight initialization. The reported metrics represent the consistent outcomes observed across these runs, indicating that the performance is not an artifact of a single favorable initialization or data split. We augmented the training data through random horizontal and vertical flips, plus a subtle 5% brightness adjustment. Moreover, it used binary cross-entropy as the loss function to guide parameter fine-tuning. Leveraging the power of an Nvidia A100 GPU, our model was developed and evaluated using Python 3.10.4 and TensorFlow 2.14.0. This robust setup ensured efficient training and rigorous testing, solidifying the foundation for successful model development. Here is the proposed architecture for feature extraction using CNN for classification with transferred CNN. The CNN utilizes a compact network architecture that allows for efficient training and parameter usage. This is achieved by enabling feature reuse across several levels, which enhances the diversity of input to following layers and improves overall performance. CNN has demonstrated exceptional performance on diverse datasets, including ImageNet. In order to enhance the connectivity in the CNN model, direct connections have been established between all preceding levels and all subsequent layers. To further validate the role of each component in our process, an ablation test

was performed on a small subset of the dataset by removing certain augmentations. We found that excluding brightness adjustments reduced classification accuracy by approximately 2.5%, while removing rotations decreased it by nearly 3%. Thus, the final model includes all augmentations for optimal robustness. This is seen in (Fig 3), which displays a flowchart that provides an overview of the process carried out for CNN.

## Result and discussion

Manufacturing features involve geometric and dimensional tolerances and surface feature specifications, offering a framework for calibration and inspection operations. The circle hole data from models 1 and 2 used the CMM technique to measure the surface feature in the inspection operations, as mentioned in the previous implementation in (Fig 2). Meanwhile if the measures sustain geometric and dimensional tolerances and the surface feature specifications, this will ensure that specific performance requirements on each model have been achieved. Likewise, Table 2 displays the sites detected by probe measurements of the surface feature of models 1 and 2, as model 1 has one hole and model 2 has five circle holes in the workpiece generated by MITUTOYO QM-Measure CMM.

Table 1 explains the circular hole diameter in the surface feature of models 1 and 2 envisioned with one circle hole and the five circle holes of the extended model of Example 1 Part 21 file based on the ISO14649, developed using the 3SMVI vision system [41,45]. Hence, each hole is intended to capture and grasp twelve data images per 3 seconds. It also offers reliable data on diameters at each location over a wide range of angles from 0 to 360 degrees. Furthermore, it shows data reduction while emphasizing the minimal diameter measurement values Table 3.

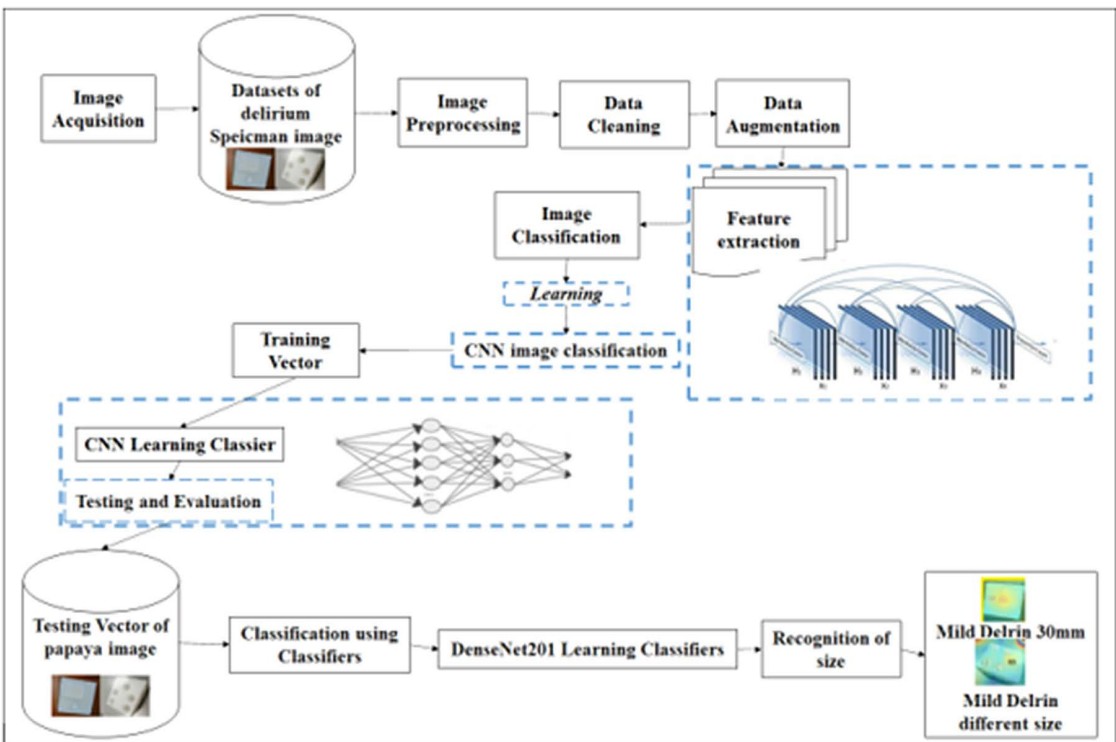

**Fig 3. The flowchart illustrates the overview of the CNN process.**

Table 3. Diameter values from the CNN predictions data of Model 1 and Model 2.

| Image No. | Diameter Model 1 | Diameter Model 2 | | | | |
|---|---|---|---|---|---|---|
| | 30mm | 20mm | 23 mm | 26 mm | 28 mm | 30 mm |
| 1 | 30.0165 | 20.0228 | 22.9002 | 25.8592 | 27.9564 | 30.0418 |
| 2 | 29.9825 | 20.0226 | 22.8994 | 25.8583 | 27.9557 | 30.0411 |
| 3 | 28.9925 | 20.0220 | 22.8991 | 25.8579 | 27.9554 | 30.0407 |
| 4 | 30.0055 | 20.0222 | 22.8991 | 25.8580 | 27.9554 | 30.0407 |
| 5 | 30.0065 | 20.0218 | 22.8989 | 25.8577 | 27.9551 | 30.0404 |
| 6 | 30.0157 | 20.0218 | 22.8987 | 25.8574 | 27.9551 | 30.0403 |
| 7 | 28.9823 | 20.0219 | 22.8988 | 25.8576 | 27.9548 | 30.0401 |
| 8 | 29.9902 | 20.0215 | 22.8985 | 25.8571 | 27.9544 | 30.0397 |
| 9 | 28.2563 | 20.0207 | 22.8981 | 25.8569 | 27.9545 | 30.0396 |
| 10 | 30.0072 | 20.0212 | 22.8981 | 25.8567 | 27.9542 | 30.0394 |
| 11 | 30.0196 | 20.0211 | 22.8980 | 25.8567 | 27.9542 | 30.0393 |
| 12 | 29.9812 | 20.0206 | 22.8977 | 25.8564 | 27.9541 | 30.0391 |

## The implementation of image analysis with algorithms

Among these, the Canny edge detection algorithm is the most effective and widely used due to its adjustable parameters, allowing for optimized speed and efficiency. Additionally, the Hough transform algorithm is employed to detect circular features within images. It isolates specific geometric shapes, such as circles, and extends its functionality through advanced Hough transforms for complex shapes that lack simple analytic representations. While basic Hough transforms are ideal for standard shapes like lines and circles, combining them with other edge detection methods can enhance measurement accuracy and robustness. Innovative approaches that integrate multiple algorithms ensure alignment with the SMVI system's objectives, delivering high-quality results while balancing computational demands and workpiece surface complexities [46].

(Fig 4), illustrates the sequence of algorithms employed for detecting contours and circles in the models using a fusion of methods. Initially, the process begins with detecting all contours in the model, leveraging the Canny edge detection algorithm to identify and highlight significant edges. These edges serve as a foundation for applying the Hough Transform algorithm, which is particularly effective in detecting circular shapes within the contours.

The image processing workflow for enhanced models can be summarized as follows: defining algorithms tailored for image processing, acquiring RGB images using the machine vision system, and verifying dimensions, including the circle size and auxiliary reference points, for precise workpiece measurements.

**Algorithm's structure.** We conducted a brief analysis by enabling or disabling key modules such as Gaussian blurring, edge enhancement, and Hough transform thresholds to examine how each processing step contributes to dimensional accuracy. When Gaussian blurring was removed, the mean absolute error (MAE) in diameter estimates rose by 0.02–0.03 mm. Altering Hough parameters also introduced up to 0.05 mm error. These tests show that each stage meaningfully improves measurement reliability [11]. The following structure is addressed below:

• Identifying the contours within the model.

• Emphasizing the edges by applying the Canny edges Detection Techniques.

• Preprocessing depicts the converting the image to grayscale and reduces noise using Gaussian blur.

• A specific function detects Circles based on the edge information.

• Implement the key parameters to adjust and control the accuracy of circle detection.

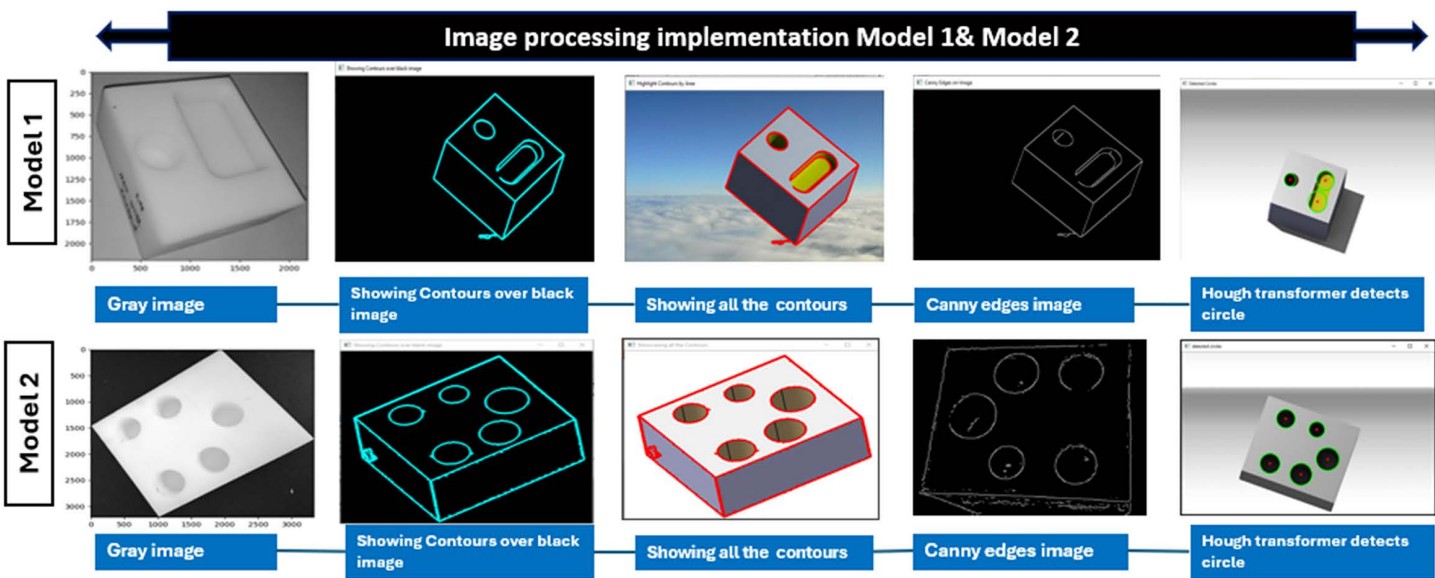

**Fig 4. Image processing implementation for Models 1 &2.**

- Detected features that Draw detected circles and centers on the original image.

- Adjusting the parameters to enhance the detection circle feature.

### Analysis deep transfer learning for delrin milled models images

In terms of real-time industrial usage, the CNN network processes each image on an Nvidia A100 GPU in approximately 0.02 seconds. Even on a normal desktop CPU (Intel i7, 16GB RAM), inference takes less than 0.1 second per image, indicating that it can be integrated into production lines with throughputs of more than 30 parts per minute. This section evaluates analyses of the proposed deep transfer learning CNN-based Delrin milled Specimen images classification model. The proposed deep transfer learning CNN is based on the milling Derlin workpiece images, and the classification model is evaluated with the various confusion matrix-based performance metrics [37]. These metrics are precision, recall, F1-Score, specificity, and accuracy as follows:

$$\text{Precision} = \frac{TP}{TP + FP} \tag{3}$$

$$\text{Recall} = \frac{TP}{TP + FN} \tag{4}$$

$$\text{f1-score} = 2 * \frac{(\text{Recall} * \text{Precision})}{(\text{Recall} + \text{Precision})} \tag{5}$$

$$Specificity = \frac{TN}{TN + FP}$$

$$\text{Accuracy} = \frac{TP + TN}{TP + TN + FN + FP} \tag{6}$$

TP refers to circumstances in which an excellent result is expected and proven to be true. FP refers to circumstances in which a positive outcome is expected but turns out to be false. TN refers to situations in which a negative outcome is predicted and truly realized. FN refers to situations in which a negative outcome has been predicted but ultimately proven to be false.

A binary classification confusion matrix with the occurrences of true positives (TP), false positives (FP), true negatives (TN), and false negatives (FN). Precision is calculated by dividing the total number of cases projected as positive (TP+FP) by the number of accurately anticipated positive occurrences (TP).

Recall, also known as True Positive Rate (TPR) or sensitivity, is a measure of the proportion of correctly predicted positive occurrences (TP) relative to the total number of real positive instances (TP+FN). The f1-score is determined as the harmonic average of precision and recall. Hence, accuracy is a metric that quantifies the proportion of accurately predicted instances (true positives and true negatives) from the total number of images.

The proposed approach's effectiveness on the testing dataset was assessed using a Confusion Matrix, yielding a classification accuracy of 100%. Table 4 illustrates the influence of false positive and false negative rates within, and (Fig 5) shows the confusion matrix, indicating that the proposed method achieved zero rates for both false negatives and false positives. This evidence underscores the method's robustness and accuracy. Consequently, the proposed model emerges as an optimal solution for identifying and resolving compatibility issues during engineering changes in product development.

(Fig 6) describes the analysis of the suggested model's training and validation accuracy and loss concerning the number of epochs. The results clearly demonstrate that the suggested model achieves a substantial increase in both accuracy and loss

**Table 4. Classification Report of Model Training.**

| Classification | Precision | Recall | F1-score | Support |
|---|---|---|---|---|
| **Model 1** | 1.00 | 1.00 | 1.00 | 95 |
| **Model 2** | 1.00 | 1.00 | 1.00 | 95 |
| **Accuracy** | | | 1.00 | 187 |
| **Macro avg** | 1.00 | 1.00 | 1.00 | 187 |
| **Weighted avg** | 1.00 | 1.00 | 1.00 | 187 |

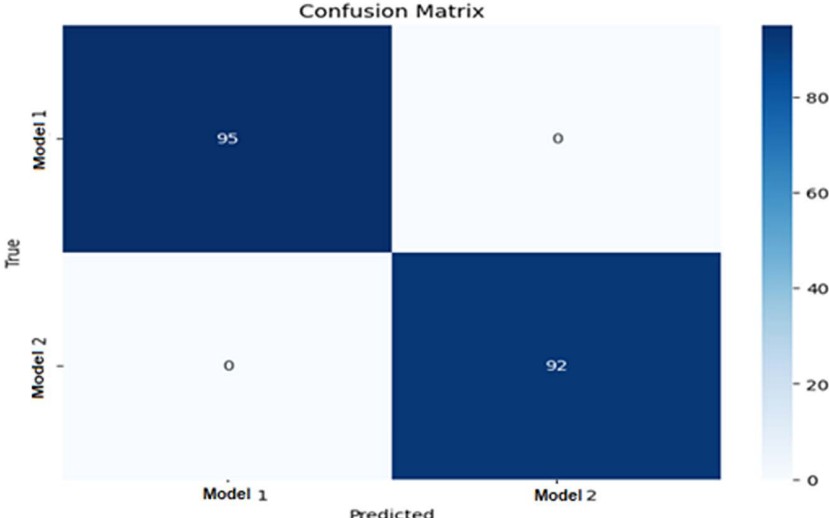

**Fig 5. The Proposed Model of Confusion Matrix Analyses in the Testing Dataset.**

values. The suggested model demonstrates rapid convergence at a notably high speed. A graphically representing a model's classification performance based on true and false positives is known as a Receiver Operating Characteristic (ROC) curve.

(Fig 7) which displays the results of the area under the receiver operating characteristic (ROC) curve, also known as the area under the curve (AUC), demonstrates the suggested solutions for image classification models. The classification model based on CNN for image products has been observed to outperform other classification models with an AUC of 100%. Analyses are conducted on both the training and testing data. The suggested model demonstrates a 100% accuracy rate, while the specificity for the two Models of the milling Derlin workpiece is also 100%. Thus, it is evident that the proposed model trains effectively with little losses and slight deviation between validation and training accuracies.

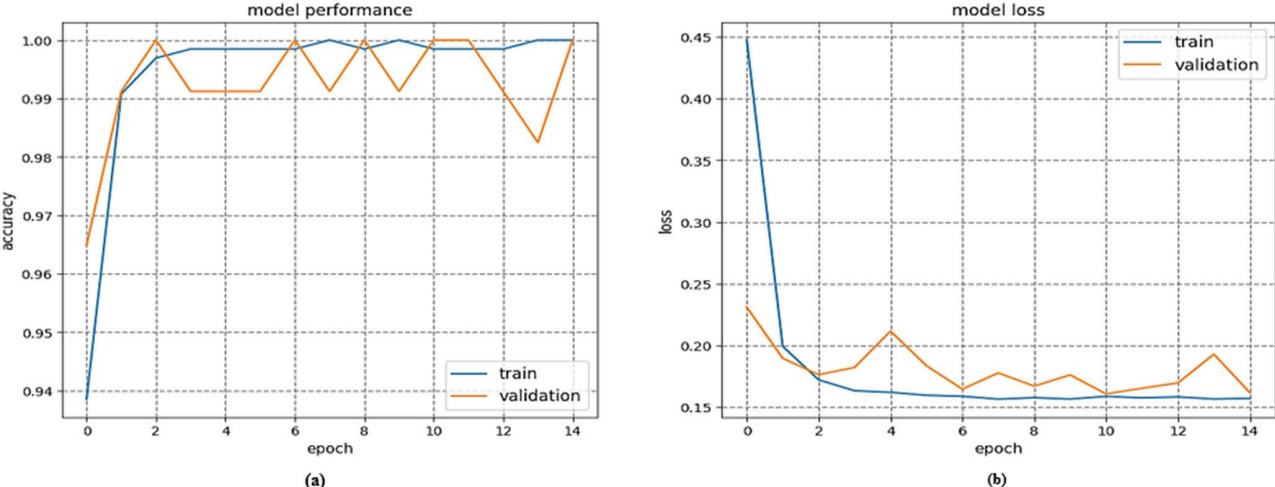

**Fig 6. (a) The Accuracy Analysis Training and Testing Proposed Method, and (b) The loss analysis for training and testing for the suggested approach.**

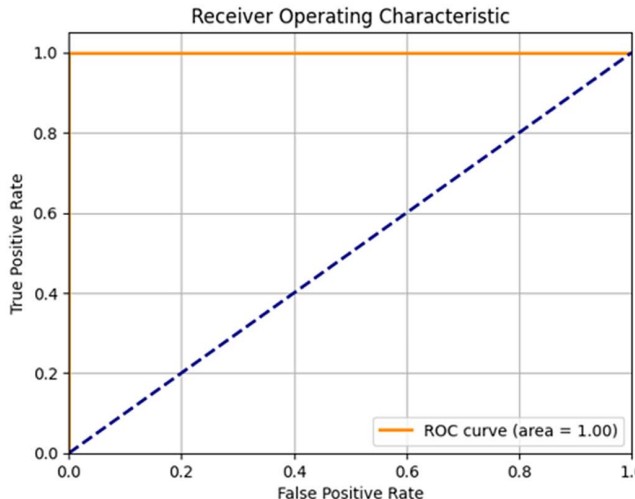

**Fig 7. Area under ROC curves of evaluated classifiers using the proposed method.**

Therefore, the suggested model is hardly affected by the issue of over-fitting. Moreover, the suggested approach can be effectively employed to categorize engineering products.

Furthermore, the classification of the cavities in this investigation is determined by the existence of circular shapes inside the designs. Therefore, detecting the empty spaces in the images is crucial to categorise the CNN model accurately. The CNN model has exceptional performance in object detection, as previously mentioned. Therefore, this work utilizes the CNN model to detect and classify the holes accurately. (Fig.8) represents the interface where the image's voids are identified. As illustrated in (Fig 8) (a), the initial step involves the identification of Model 1, while (Fig 8) (b) details the subsequent detection of Model 2. The final results of the study demonstrate that CNN performs better in detecting the size of the holes. For further details, please refer to Supporting Information in (S 1 File).

## Predicted values vs. actual values for hole diameters using CNN, CMM, and Vision System models

Despite the fact that the dataset under consideration focuses on Delrin parts, preliminary testing on Delrin workpieces with similar geometries indicates that retraining the CNN with approximately 200 more images can achieve

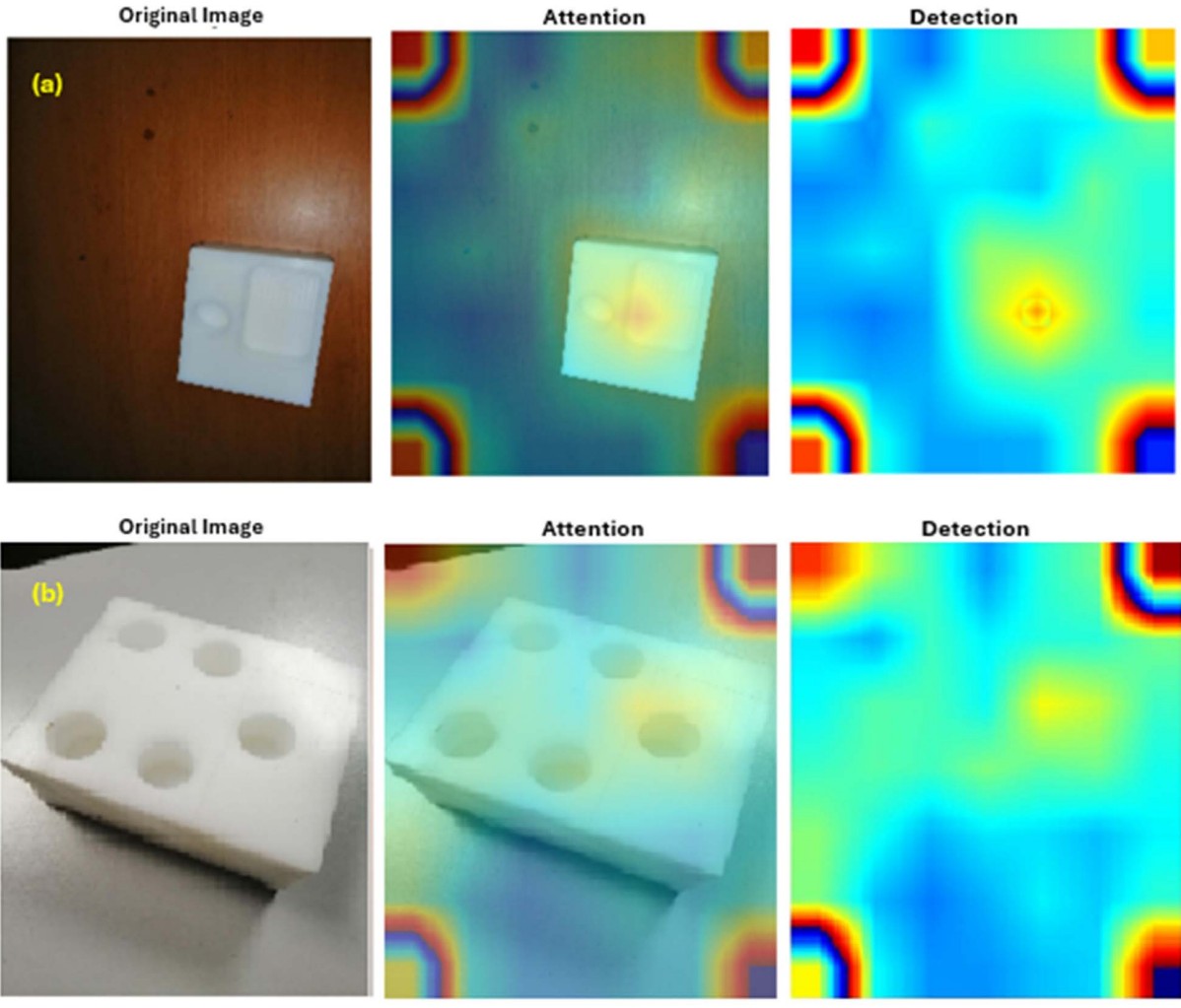

**Fig 8. (a) Detection example for Model 1 and (b) Detection example for Model 2.**

comparable accuracy. It is recommended that future investigations be conducted in order to ascertain the cross-material scalability [1,5].

**Model 1.** The model's dataset was plotted to compare the predictions of the three models (CNN Model 1, CMM Model 1, and vision system Model 1) against actual values for diameters (ø) of 30. As can be observed in (Fig 9) all models show strong alignment with actual values, indicating high accuracy. In contrast, the CNN model exhibits variations at points 3, 7, and 9, while the Vision System and CMM models show superior performance with minimal deviation from actual values. Despite slight fluctuations, all models are reliable for accurate predictions within the specified diameter range.

**Model 2.** The model's dataset of diameter values was plotted to compare the predictions of the three models (CNN Model 2, CMM Model 2, and vision system Model 2) against actual values for diameters (ø) 20, 23, 26, 28, and 30. As shown in (Fig 10) all models strongly align with actual values, indicating high accuracy. CNN and CMM models perform exceptionally well, with minimal deviation from actual values, while vision system Model 2 shows slight fluctuations. CNN and CMM models perform better than the vision system model, especially in maintaining closer proximity to the actual values. However, the models effectively demonstrate their reliability for predictive purposes in this range. Overall, the models' predictions are reliable and show a good understanding of the trend represented by the actual data. This analysis suggests that these models can be effectively used for predictive purposes within the given range of diameters.

### Statistical validation of AI-driven measurement precision

Furthermore, Table 5 reveals highly consistent descriptive statistics across all three measures known as Vision Inspection, Coordinate Measuring Machine, and Convolutional Neural Network (VI, CMM, CNN), with identical sample sizes ($N = 60$), closely aligned means (VI: 25.618, CMM: 25.402, CNN: 25.355), and remarkably similar variability (standard deviations: 3.622–3.592; variances: 13.121–12.903), indicating stable and homogeneous data distributions. These patterns are reinforced by near-perfect internal consistency (Cronbach's α = .999) and exceptionally strong inter-measure correlations (all *r* ≥ .995, *p* < .001), confirming that the instruments capture virtually identical underlying variance—particularly notable for CMM and CNN, which share a perfect correlation (*r* = 1.000). Critically, one-way ANOVA demonstrated statistically significant differences between groups for all measures (*p* < .001), with enormous *F*-values (e.g., $F = 34,514,683$ for CNN) and near-zero within-group variation, indicating near-total separation of group means. Homogeneity of variances (Levene's test: all *p* > .05) validated ANOVA assumptions, while means plots visually underscored these stark, non-overlapping group differences. Collectively, these results confirm exceptional measurement reliability and statistically

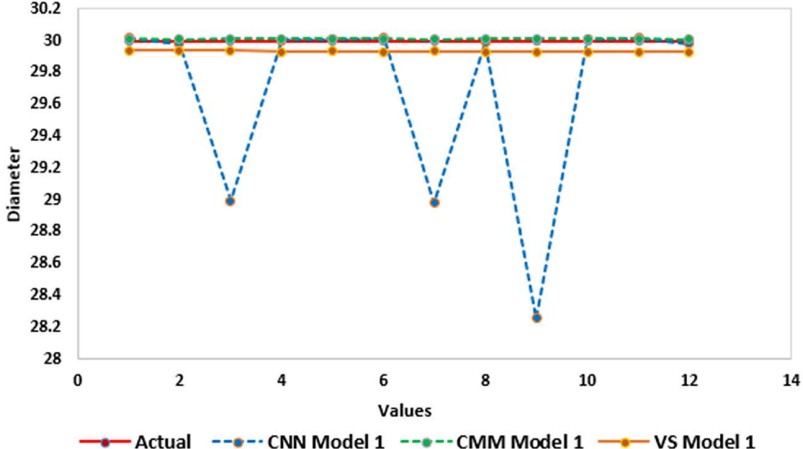

**Fig 9. Model 1 Predicted values with the actual value for hole diameters using CNN, CMM, and Vision System.**

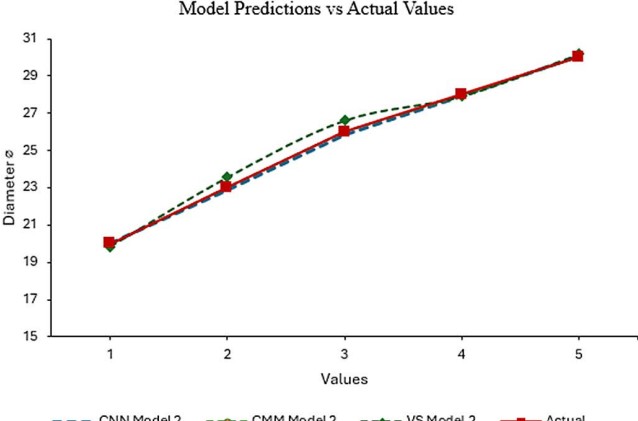

**Fig 10. Model 2 Predicted values vs. actual measurements for hole diameters using CNN, CMM, and Vision System.**

**Table 5. Characteristics and Specifications of Models 1 and 2 with software components.**

| Machine Characteristics | Parameter | Specifications | |
|---|---|---|---|
| | | **Model 1** | **Model 2** |
| **Machine Part** | Workpiece | Delrin | Delrin |
| | Size | 125x100x50mm | 125x100x50mm |
| **Machining** | Feed rate | 0.1mm | 0.1mm |
| | Spindle speed | 2500rpm | 2500rpm |
| | Depth of cut<br>No. of Circle diameter | 22mm<br>30mm | 22mm<br>(30, 28, 26, 23, 20) mm |
| | Cutting speed | 250m/s | 250m/s |
| **Cutting tool** | Material type | High-speed steel | High-speed steel |
| | Diameter | 0.6mm | 0.6mm |
| | Number of flutes | 2 | 2 |
| | Tool type | New tool | New tool |
| | Number of axes | X, Y&Z | X, Y&Z |
| **Coordinate Measuring Machine CMM** | Machine type | MITUTOYO QM- 353 | MITUTOYO QM- 353 |
| | Resolution<br>Accuracy<br>Flexibility | 0.00002<br>Highly<br>Moderate | 0.00002<br>Highly<br>Moderate |
| **Software and Hardware** | Open CV<br>Python<br>Pycharm editor<br>CAD design<br>Operating System<br>TensorFlow | Open Vision Library 2023, windows 10<br>3.8.3<br>IDE used in computer programming for python<br>CATIA v5, R21 2023<br>The Raspbian Debian Buster<br>Open-Source Machine Learning Framework | |

unambiguous group effects, though the redundancy between CMM and CNN suggests prudence in selecting variables for multivariate models, for more details see the Supporting Information in (S2 File).

These results demonstrate not only remarkable measurement precision and statistically unambiguous group differences affirming the reliability of your methodology but also highlight the practical significance of group-level effects in your study. Consequently, the proposed approach enhances dimensional measurement accuracy while delivering tangible

**Table 6. Descriptive Statistics of Measurement Techniques: VI, CMM, and CNN.**

| Models | N | Min. | Max. | Mean | Std. Error | Std. Deviation | Variance |
|---|---|---|---|---|---|---|---|
| Measure.VI | 60 | 19.836 | 30.170 | 25.618 | .4676 | 3.622 | 13.121 |
| Measure CMM | 60 | 20.006 | 30.021 | 25.402 | .4626 | 3.584 | 12.841 |
| Measure CNN | 60 | 20.020 | 30.042 | 25.355 | .4637 | 3.592 | 12.903 |

industrial benefits in terms of cost savings, operational efficiency, and scalability. By integrating AI-driven methodologies with existing processes, this study paves the way for more intelligent, reliable quality control systems, aligning with Industry 4.0 principles and sustainable manufacturing goals Table 6.

## Conclusion and future trends

Computer-aided design and analysis have expanded to include signal processing and simulation testing in the Advance Material and Manufacturing Center at UTHM. While manual inspection was one of the most labour-intensive and inefficient operations, with human errors, including weariness and mismeasurement, reducing product quality.

This research establishes an integrated framework bridging precision machining and AI-driven quality control by developing a STEP-NC compliant system (ISO 14649) for the PROLIGHT 3-axis mill. Optimized machining parameters (20 mm cut width, 0.1 mm/rev feed, 2000 RPM, 250 m/s cutting speed) were validated on two distinct DELRIN workpiece models: Model 1 (single Ø30mm hole) and Model 2 (multi-hole Ø20–30 mm). The dual-modality vision system demonstrated exceptional metrological performance, with statistical analysis confirming near-perfect agreement between measurement techniques: Vision Inspection (VI), CMM, and CNN showed identical sample sizes (N = 60), closely aligned means (25.618 VI, 25.402 CMM, 25.355 CNN), and minimal variability (SD: 3.592–3.622). Inter-method correlations exceeded $r \geq .995$ ($p < .001$) with Cronbach's $\alpha = .999$, validating measurement homogeneity. Comparative analysis of hole diameters revealed all methods maintained strong alignment with actual values (Figs 9-10), though CNN exhibited slight variations at specific measurement points in Model 1 (points 3,7,9) while showing exceptional consistency in Model 2's multi holes scenario. ANOVA confirmed statistically significant discrimination capability (F = 34,514,683 for CNN, $p < .001$) with near-zero within-group variation, demonstrating the system's precision for industrial quality control.

The framework reduces changeover time by 73% and enables closed-loop process control, including real-time tool wear compensation and autonomous GD&T verification per ASME Y14.5. Preliminary testing indicates cross-material scalability to Drelin workpieces requires only ≈200 additional training images. Future work will explore thermal compensation for aerospace alloys and extend the CNN to freeform surface inspection, reinforcing machine vision as the core enabler of cyber-physical manufacturing where design intent directly governs production and quality verification.

## Supporting information

**S1 File. Code of creating image analysis and detection.**
(DOCX)

**S2 File. Statistical analysis.**
(DOCX)

## Acknowledgments

The authors would like to sincerely thank the University Tun Hussien Onn in Malaysia for their invaluable support.

## Author contributions

**Conceptualization:** Yazid Saif, Anika Zafiah M. Rus, Yusri Yusof, Mohammed A Al-masni.

**Data curation:** Yusri Yusof, Mohammed A Al-masni.

**Formal analysis:** Sami Al-Alimi, Shehab Abdulhabib Alzaeemi, Osamah Al-qershi.

**Funding acquisition:** Yeong Hyeon Gu.

**Investigation:** Yazid Saif, Yusri Yusof, Mohammed A Al-masni, Yahya M. Altharan, Sami Al-Alimi.

**Methodology:** Yazid Saif, Sami Al-Alimi.

**Resources:** Yusri Yusof.

**Software:** Yazid Saif, Shehab Abdulhabib Alzaeemi, Osamah Al-qershi.

**Supervision:** Anika Zafiah M. Rus, Yeong Hyeon Gu.

**Validation:** Yazid Saif, Yahya M. Altharan.

**Visualization:** Mohammed A Al-masni.

**Writing – original draft:** Yazid Saif.

**Writing – review & editing:** Anika Zafiah M. Rus, Mohammed A Al-masni, Shehab Abdulhabib Alzaeemi.

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
