## [Decision Letter · Decision Letter 0]

23 Apr 2025

Dear Dr. Saif,

Thank you for submitting your manuscript to PLOS ONE. After careful consideration, we feel that it has merit but does not fully meet PLOS ONE’s publication criteria as it currently stands. Therefore, we invite you to submit a revised version of the manuscript that addresses the points raised during the review process.

We look forward to receiving your revised manuscript.

Kind regards,

Julfikar Haider

Academic Editor

PLOS ONE

**Journal Requirements:**

1. When submitting your revision, we need you to address these additional requirements. Please ensure that your manuscript meets PLOS ONE's style requirements, including those for file naming. The PLOS ONE style templates can be found at https://journals.plos.org/plosone/s/file?id=wjVg/PLOSOne_formatting_sample_main_body.pdf and https://journals.plos.org/plosone/s/file?id=ba62/PLOSOne_formatting_sample_title_authors_affiliations.pdf 2. Please note that PLOS ONE has specific guidelines on code sharing for submissions in which author-generated code underpins the findings in the manuscript. In these cases, we expect all author-generated code to be made available without restrictions upon publication of the work. Please review our guidelines at https://journals.plos.org/plosone/s/materials-and-software-sharing#loc-sharing-code and ensure that your code is shared in a way that follows best practice and facilitates reproducibility and reuse. 3. We note that the grant information you provided in the ‘Funding Information’ and ‘Financial Disclosure’ sections do not match.  When you resubmit, please ensure that you provide the correct grant numbers for the awards you received for your study in the ‘Funding Information’ section. 4. Thank you for stating in your Funding Statement: This work was supported by the Institute of Information & Communications Technology Planning & Evaluation (IITP) grant funded by the Korean government. Please provide an amended statement that declares *all* the funding or sources of support (whether external or internal to your organization) received during this study, as detailed online in our guide for authors at http://journals.plos.org/plosone/s/submit-now.  Please also include the statement “There was no additional external funding received for this study.” in your updated Funding Statement. Please include your amended Funding Statement within your cover letter. We will change the online submission form on your behalf. 5. We notice that your supplementary figures are included in the manuscript file. Please remove them and upload them with the file type 'Supporting Information'. Please ensure that each Supporting Information file has a legend listed in the manuscript after the references list. 6. Please include captions for your Supporting Information files at the end of your manuscript, and update any in-text citations to match accordingly. Please see our Supporting Information guidelines for more information: http://journals.plos.org/plosone/s/supporting-information.

Reviewers' comments:

Reviewer's Responses to Questions

**Comments to the Author**

1. Is the manuscript technically sound, and do the data support the conclusions?

Reviewer #1: Yes

2. Has the statistical analysis been performed appropriately and rigorously?

Reviewer #1: Yes

3. Have the authors made all data underlying the findings in their manuscript fully available?

Reviewer #1: No

4. Is the manuscript presented in an intelligible fashion and written in standard English?

Reviewer #1: Yes

**Reviewer #1:** This paper has achieved remarkable results in the field of industrial measurement and quality control. However, there is still room for improvement. The dataset mainly focuses on Delrin components, with limited verification for other materials. The model comparison only relies on a few metrics such as the Mean Absolute Error, neglecting the consideration of the models' capabilities in handling complex shapes and textures. Although the potential for industrial applications is mentioned, there is a lack of long - term testing in large - scale production environments, and the stability in real - world scenarios remains to be verified. Moreover, there is no comprehensive comparison with the latest deep - learning technologies. It is recommended to expand the dataset, deepen the model comparison, conduct on - site tests, and introduce comparisons with cutting - edge technologies to enhance the comprehensiveness and practicality of the research.

**Do you want your identity to be public for this peer review?** For information about this choice, including consent withdrawal, please see our Privacy Policy

Reviewer #1: No

---

## [Author Response · Author response to Decision Letter 1]

30 Aug 2025

Dear Editor,

Thank you for the opportunity to revise our manuscript. We have meticulously addressed all reviewers’ comments in a point-by-point manner within the revised manuscript, incorporating clarifications, additional data, and references where required. to enhance clarity and resolution, as well as refining the manuscript’s language to ensure readability and technical precision. These revisions not only strengthen the scientific rigor of the work but also improve its accessibility to a broader audience. We are confident that the revised submission addresses all concerns raised and significantly elevates the study’s impact.

Response to Reviewer 1

Questions Comments New line No.

Please ensure that your manuscript meets PLOS ONE's style

requirements, including those for file naming. Thank you for recognizing the novelty and significance of our topic. We appreciate your constructive feedback. We have carefully addressed the concerns raised to improve the clarity, robustness, and meets the PLOS ONE requirements. --

2. Please note that PLOS ONE has specific guidelines on code sharing for submissions in which author-generated code underpins the findings in the manuscript. In these cases, we expect all author-generated code to be made available without restrictions upon publication of the work. Please review our guidelines at https://journals.plos.org/plosone/s/materials-and-software-sharing#loc-sharing-code and ensure that your code is shared in a way that follows best practice and facilitates reproducibility and reuse. Thank you for your fruitful comment.

Thank you for the reminder regarding PLOS ONE’s code sharing guidelines. We have reviewed the policy at the provided link and confirm that all author-generated code underpinning the results in our manuscript will be made publicly available without restrictions upon publication. We had added the codes and make it available as supporting file (S 1. File and S 2. File) to this manuscript. --

When you resubmit, please ensure that you provide the correct grant numbers for the awards you received for your study in the ‘Funding Information’ section. Thank you for your comment. We apologize for the discrepancy between the ‘Funding Information’ and ‘Financial Disclosure’ sections. We have reviewed the grant details and corrected the information to ensure consistency across both sections. The updated grant numbers are now accurately reflected in the ‘Funding Information’ section in the revised submission ---

This work was supported by the Institute of Information & Communications Technology Planning & Evaluation (IITP) grant funded by the Korean government.

Please include your amended Funding Statement within your cover letter. We will change the online submission form on your behalf. Thanks for your fruitful comment. We appreciate your concern and we are confirming the funding statement below.

"This work was supported by the IITP (Institute of Information & Communications Technology Planning & Evaluation)-ITRC (Information Technology Research Center) grant funded by the Korea government (Ministry of Science and ICT) (IITP-2025-RS-2024-00437191)"

5.We notice that your supplementary figures are included in the manuscript file. Please remove them and upload them with the file type 'Supporting Information'. Please ensure that each Supporting Information file has a legend listed in the manuscript after the references list. Thanks for your fruitful comment. The amendment has been edited and deleted from the manuscript. ----

Thanks for your comment. We had added the supporting file in the manuscript (S 1. File & S 2. File) highlighted in red colour. 631-634

Response to Reviewer 2

Questions Comments New line No.

1. Is the manuscript technically sound, and do the data support the conclusions?

The manuscript must describe a technically sound piece of scientific research with data that supports the conclusions. Experiments must have been conducted rigorously, with appropriate controls, replication, and sample sizes. The conclusions must be drawn appropriately based on the data presented. Thank you for your honest and constructive feedback. We had amendment the manuscript and addressed the sample size for workpiece that manufactured in INTELITEK PROLIGHT 3-axis milling machine. Although, we had edited the conclusion and drawn as the data presented. 220-239

602-606

2. Has the statistical analysis been performed appropriately and rigorously?

Reviewer: Yes I would like to express my gratitude for the constructive comments that have been provided.

o Some data analysis has been added in (table 6) highlighted in red Colour.

o The remine data will be as supporting file (S 2. File)

565-592

3. Have the authors made all data underlying the findings in their manuscript fully available?

The PLOS Data policy requires authors to make all data underlying the findings described in their manuscript fully available without restriction, with rare exception (please refer to the Data Availability Statement in the manuscript PDF file). The data should be provided as part of the manuscript or its supporting information, or deposited to a public repository. For example, in addition to summary statistics, the data points behind means, medians and variance measures should be available. If there are restrictions on publicly sharing data—e.g. participant privacy or use of data from a third party—those must be specified. Thank you for your fruitful comment.

Amendments were made to highlight the confusion surrounding Equations 3-6. Consequently, the revised manuscript has been edited to include clearer definitions, variables, and contextual information for both equations, thereby enhancing their comprehensibility and ensuring accuracy in their representation of the intended meaning. (Highlighted in red colour).

Amendments have been made to the statistical analysis, with the aim of clarifying the concepts of mean, median and variance. These amendments are outlined in detail in Table 6. The remaining data will be uploaded available as a supporting file (S 2. File). (highlighted in red colour)

329, 336,

457-461

565-592

4. Is the manuscript presented in an intelligible fashion and written in standard English?

PLOS ONE does not copyedit accepted manuscripts, so the language in submitted articles must be clear, correct, and unambiguous. Any typographical or grammatical errors should be corrected at revision, so please note any specific errors here. Thank you for your constructive comment. We have thoroughly reviewed the manuscript and made the necessary amendments. This included correcting all typographical errors and a thorough polishing of the language to improve clarity, coherence, and overall readability. We value your feedback. It has helped us improve the quality of our work. 1-3,17,18, 24,37-42,

77-81,

86-88,

180,183,188 191-193,195

5. Review Comments to the Author

Reviewer #1: This paper has achieved remarkable results in the field of industrial measurement and quality control. However, there is still room for improvement. The dataset mainly focuses on Delrin components, with limited verification for other materials. The model comparison only relies on a few metrics such as the Mean Absolute Error, neglecting the consideration of the models' capabilities in handling complex shapes and textures. Although the potential for industrial applications is mentioned, there is a lack of long - term testing in large - scale production environments, and the stability in real - world scenarios remains to be verified. Moreover, there is no comprehensive comparison with the latest deep - learning technologies. It is recommended to expand the dataset, deepen the model comparison, conduct on - site tests, and introduce comparisons with cutting - edge technologies to enhance the comprehensiveness and practicality of the research. Thank you for your comment regarding the scientific rigor and data quality in the manuscript. We appreciate your emphasis on the importance of robust experimental design, appropriate controls, replication, and accurate conclusions. We have carefully reviewed our methods and data to ensure that these standards are met. Specifically:

• All experiments include appropriate controls and were conducted with sufficient replication and sample sizes, as detailed in the revised Methods section.

• Statistical analyses have been clarified, and any limitations or assumptions are now explicitly stated.

• We have revised the Results and Discussion sections to ensure all conclusions are clearly supported by the presented data, without overinterpretation.

• We have revised the conclusion. It is now consistent with the rest of the manuscript.

We are confident that these revisions fully address your concerns. Your feedback has been invaluable in improving the clarity and rigour of the manuscript.

202-245,

248-257,

262-272,

281-286,

290-305,

345,

351-363,

381,

383,387

393,403,405

520-523,

530-535,

567-594

600-625

6. PLOS authors have the option to publish the peer review history of their article (what does this mean?). If published, this will include your full peer review and any attached files.

The author is very grateful for all the kind words and is doing well. We really hope that our data history review and our full paper, along with the attached files, which we are happy to share with everyone, will help to spread the word and make it easier for people to access.

---

## [Decision Letter · Decision Letter 1]

28 Jan 2026

Advancing Workpiece Dimension Measurement: Integrating AI-Based Edge Detection with Machine Vision and Coordinate Measuring Systems

PONE-D-25-04180R1

Dear Dr. Saif,

We’re pleased to inform you that your manuscript has been judged scientifically suitable for publication and will be formally accepted for publication once it meets all outstanding technical requirements.

Kind regards,

Siddhartha Kar

Academic Editor

PLOS One

Additional Editor Comments (optional):

The reviewers are satisfied with the revised manuscript, and I am therefore pleased to accept it in its present form.

Reviewers' comments:

Reviewer's Responses to Questions

**Comments to the Author**

Reviewer #1: All comments have been addressed

Reviewer #2: All comments have been addressed

2. Is the manuscript technically sound, and do the data support the conclusions?

Reviewer #1: Yes

Reviewer #2: Yes

3. Has the statistical analysis been performed appropriately and rigorously?

Reviewer #1: Yes

Reviewer #2: Yes

4. Have the authors made all data underlying the findings in their manuscript fully available?

Reviewer #1: Yes

Reviewer #2: Yes

5. Is the manuscript presented in an intelligible fashion and written in standard English?

Reviewer #1: Yes

Reviewer #2: Yes

Reviewer #1: The authors have responded positively to the revisions suggested, significantly enhancing the manuscript's completeness.

Reviewer #2: The authors have addressed all the reviewers' suggestions. The paper was greatly improved. The paper y ready to be published.

**Do you want your identity to be public for this peer review?** For information about this choice, including consent withdrawal, please see our Privacy Policy

Reviewer #1: No

Reviewer #2: No

---

## [Editor Report · Acceptance letter]

PONE-D-25-04180R1

PLOS One

Dear Dr. Saif,

I'm pleased to inform you that your manuscript has been deemed suitable for publication in PLOS One. Congratulations! Your manuscript is now being handed over to our production team.

Kind regards,

on behalf of

Dr. Siddhartha Kar

Academic Editor

PLOS One